# Modeling the Density of Pixel-level Self-supervised Embeddings for Unsupervised Pathology Segmentation in Medical CT

**Mikhail Goncharov**[1,*,†]  **Eugenia Soboleva**[2,*]  **Daniil Ignatyev**[3]  **Mariia Donskova**[3]
**Mikhail Belyaev**[3]  **Ivan Oseledets**[1]  **Marina Munkhoeva**[1,4]  **Maxim Panov**[5]

[1]AXXX, Moscow, Russia  [2]Yandex  [3]AUMI AI  [4]Lomonosov Moscow State University
[5]MBZUAI

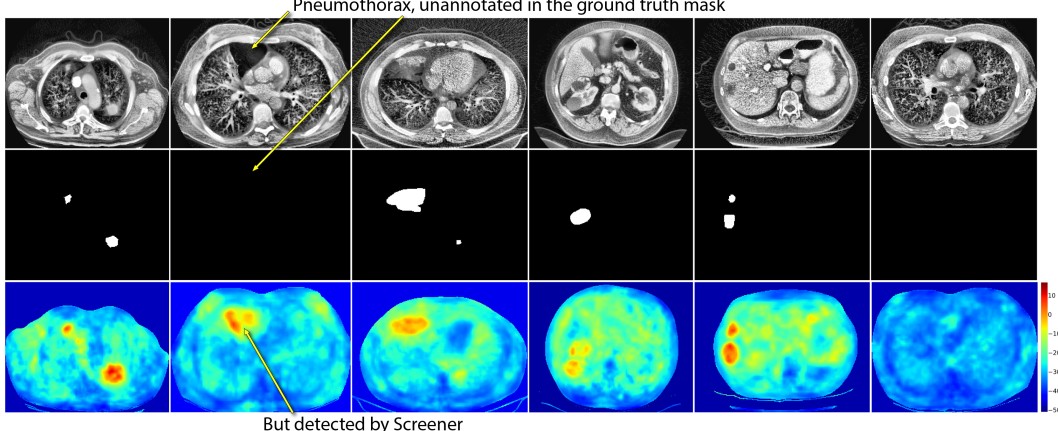

Figure 1: Examples of CT image slices (the first row), the ground truth pathology masks (the second row) and the anomaly maps predicted by our *fully self-supervised* Screener model (the third row).

## Abstract

Accurate detection of all pathological findings in 3D medical images remains a significant challenge, as supervised models are limited to detecting only the few pathology classes annotated in existing datasets. To address this, we frame pathology detection as an unsupervised visual anomaly segmentation (UVAS) problem, leveraging the inherent rarity of pathological patterns compared to healthy ones. We enhance the existing density-based UVAS framework with two key innovations: (1) dense self-supervised learning for feature extraction, eliminating the need for supervised pretraining, and (2) learned, masking-invariant dense features as conditioning variables, replacing hand-crafted positional encodings. Trained on over 30,000 unlabeled 3D CT volumes, our fully self-supervised model, Screener, outperforms existing UVAS methods on four large-scale test datasets comprising 1,820 scans with diverse pathologies. Furthermore, in a low-shot supervised fine-tuning setting, Screener surpasses existing self-supervised pretraining methods, establishing it as a state-of-the-art foundation for pathology segmentation. The code and pretrained models are available at
https://github.com/mishgon/screener.

## 1 Introduction

Accurate identification, localization, and classification of *all* pathological findings in 3D medical images remain a significant challenge in medical computer vision. While supervised models have

---

[*]Equal contribution.
[†]Corresponding author: mig0nch@yandex.ru.

shown promise, their utility is limited by the scarcity of labeled datasets, which often contain annotations for only a few pathologies. For example, Figure 1 shows 2D slices of 3D computed tomography (CT) images (first row) from public datasets providing annotations of lung cancer, pneumonia, kidney tumors, or liver tumors, while annotations of other pathologies, e.g., pneumothorax, are missing. This restricts supervised models to narrow, task-specific applications.

However, unlabeled CT datasets are abundant but often remain unused. Leveraging these datasets, we aim to develop an unsupervised model capable of distinguishing pathological regions from normal ones. Our core assumption is that pathological patterns are statistically rarer than healthy patterns in CT images. This frames pathology segmentation as an unsupervised visual anomaly segmentation (UVAS) problem.

Although existing UVAS methods have been extensively explored for natural images, their adaptation to medical imaging is challenging. One obstacle is that uncurated CT datasets include many patients with pathologies, and there is no automatic way to filter them out to ensure a training set composed entirely of normal (healthy) images — a common requirement for synthetic-based (Zavrtanik et al., 2021; Marimont & Tarroni, 2023) and reconstruction-based (Baur et al., 2021; Schlegl et al., 2019) UVAS methods. Density-based approaches (Gudovskiy et al., 2022; Zhou et al., 2024) are better suited, as they model image patterns probabilistically and assume abnormal patterns are rare rather than absent. To model the density of image patterns, existing methods encode them into feature maps using an ImageNet-pretrained encoder. Therefore, their performance on medical images degrades due to a domain shift. Supervised medical encoders like STU-Net (Huang et al., 2023) might seem viable, but our experiments show they also underperform, likely because their features are too specific and lack discriminative information for pathology segmentation.

To address these challenges, we propose using dense self-supervised learning (SSL) (O. Pinheiro et al., 2020) to pretrain more discriminative feature maps of CT images and employ them in the density-based UVAS framework. Thus, our model learns the distribution of dense SSL embeddings and assigns high anomaly scores to image regions where embeddings fall into low-density regions.

Inspired by dense SSL, we also generalize the idea of conditioning in density-based UVAS methods. Existing works (Gudovskiy et al., 2022; Zhou et al., 2024) use hand-crafted conditioning variables such as pixel-wise sinusoidal positional embeddings. We replace them by *learned* pixel-wise contextual embeddings capturing global characteristics of individual image regions, e.g. their anatomical position, patient's age, etc. At the same time, we eliminate local information about presence of pathologies from the learned conditioning variables by enforcing their invariance to image masking.

We train our model, *Screener*, on 30,000 unlabeled CT volumes and evaluate it on 1,820 scans in two settings. First, as a fully unsupervised model, it achieves remarkable results (Figure 1), significantly outperforming existing UVAS methods. Second, after fine-tuning for downstream pathology segmentation tasks, Screener rivals other state-of-the-art pretrained models.

Our key contributions are four-fold:

- **Dense self-supervised features for density-based UVAS.** We demonstrate that dense self-supervised representations can be successfully used and even preferred over supervised feature extractors in density-based UVAS methods. This enables a novel fully self-supervised UVAS framework for domains with limited labeled data.

- **Learned conditioning variables.** We propose novel self-supervised conditioning variables for density-based UVAS, simplifying the conditional distributions and enabling a simple Gaussian density model to perform on par with normalizing flows.

- **State-of-the-art UVAS results in CT.** This work presents the first large-scale evaluation of UVAS methods for CT images, showing state-of-the-art performance on unsupervised semantic segmentation of pathologies in diverse anatomical regions, including lung cancer, pneumonia, liver and kidney tumors.

- **State-of-the-art pretraining for pathology segmentation.** We introduce a novel pretraining method that distills Screener into a UNet, enabling supervised fine-tuning and matching the performance of state-of-the-art self-supervised pretraining methods.

## 2 BACKGROUND & NOTATION

### 2.1 DENSITY-BASED UVAS

The core idea of density-based UVAS methods is to assign high anomaly scores to image regions containing *statistically rare* patterns. To implement this idea, they involve two models, which we call a *descriptor model* and a *density model*. The descriptor model encodes image patterns into vector representations, while the density model learns their distribution and assigns anomaly scores.

The descriptor model $f_{\theta^{\text{desc}}}$ is usually a pretrained fully-convolutional neural network. For a 3D image $\mathbf{x} \in \mathbb{R}^{H \times W \times S}$, it produces feature maps $\mathbf{y} \in \mathbb{R}^{h \times w \times s \times d^{\text{desc}}}$ consisting of vectors $\mathbf{y}[p] \in \mathbb{R}^{d^{\text{desc}}}$, which we call *descriptors* of positions $p \in P = \{1, \ldots, h\} \times \{1, \ldots, w\} \times \{1, \ldots, s\}$.

The density model $q_{\theta^{\text{dens}}}(y)$ estimates the descriptors' marginal density $q_Y(y)$ (here, $Y$ denotes the descriptor of a random position in a random image). For an abnormal pattern at position $p$, the descriptor $\mathbf{y}[p]$ is expected to lie in a low-density region, resulting in a low $q_{\theta^{\text{dens}}}(\mathbf{y}[p])$. Conversely, normal patterns correspond to high density values. During inference, the negative log-density values, $-\log q_{\theta^{\text{dens}}}(\mathbf{y}[p])$, are used as anomaly segmentation scores.

This framework can be extended with a conditioning mechanism. For each position $p$, an auxiliary variable $\mathbf{c}[p] \in \mathbb{R}^{d^{\text{cond}}}$, called a *condition*, is introduced. Instead of modeling the marginal density $q_Y(y)$, the conditional density $q_{Y|C}(y \mid c)$ is learned for each condition $c$, where $(Y, C)$ represents the descriptor and condition at a random position in a random image. At inference, the negative log-conditional densities, $-\log q_{\theta^{\text{dens}}}(\mathbf{y}[p] \mid \mathbf{c}[p])$, serve as anomaly scores. State-of-the-art methods (Gudovskiy et al., 2022; Zhou et al., 2024) follow this conditional framework using sinusoidal positional encodings as conditions.

### 2.2 DENSE JOINT EMBEDDING SSL

Joint embedding SSL models learn meaningful image embeddings by generating positive pairs — augmented views of the same image (e.g., random crops). They optimize embeddings to capture mutual information between views, making them both discriminative (distinguishing images) and augmentation-invariant (predictable across views). Contrastive methods, e.g., SimCLR (Chen et al., 2020), explicitly push apart embeddings of different images, while non-contrastive methods, e.g., VICReg (Bardes et al., 2021), avoid embeddings' collapse through regularization. Details on SimCLR and VICReg are in the Appendix B.

*Dense* SSL methods extend this idea to learn image feature maps consisting of pixel-wise embeddings that encode information about different spatial positions in the image. To this end, they define positive pairs at the pixel level: two embeddings are positive if they correspond to the same absolute position in the original image, but are predicted from different augmented crops (see the upper part of Figure 2 for illustration). Thus, dense SSL enforces feature maps to be equivariant w.r.t. crops, while encouraging dissimilarity between embeddings from different positions. DenseCL (Wang et al., 2021) and VADER (O. Pinheiro et al., 2020) use contrastive losses, while VICRegL (Bardes et al., 2022) adopts a VICReg objective.

## 3 METHOD

**Novelty statement.** Our method, illustrated in Figure 2, enhances the density-based UVAS framework with two key innovations. First, instead of relying on generic backbones, we *pretrain our descriptor model via dense SSL* which enables domain-specific, high-resolution, customizable and more discriminative descriptors (Section 3.1). Second, we introduce novel *masking-invariant conditioning variables, also learned via dense SSL* (Section 3.2), and largely simplifying further conditional density modeling (Section 3.3). Beyond these contributions, we distill the overall UVAS inference pipeline to a single UNet architecture, which makes it suitable for further supervised fine-tuning. This allows us to reinterpret our framework as a *novel self-supervised pretraining method*.

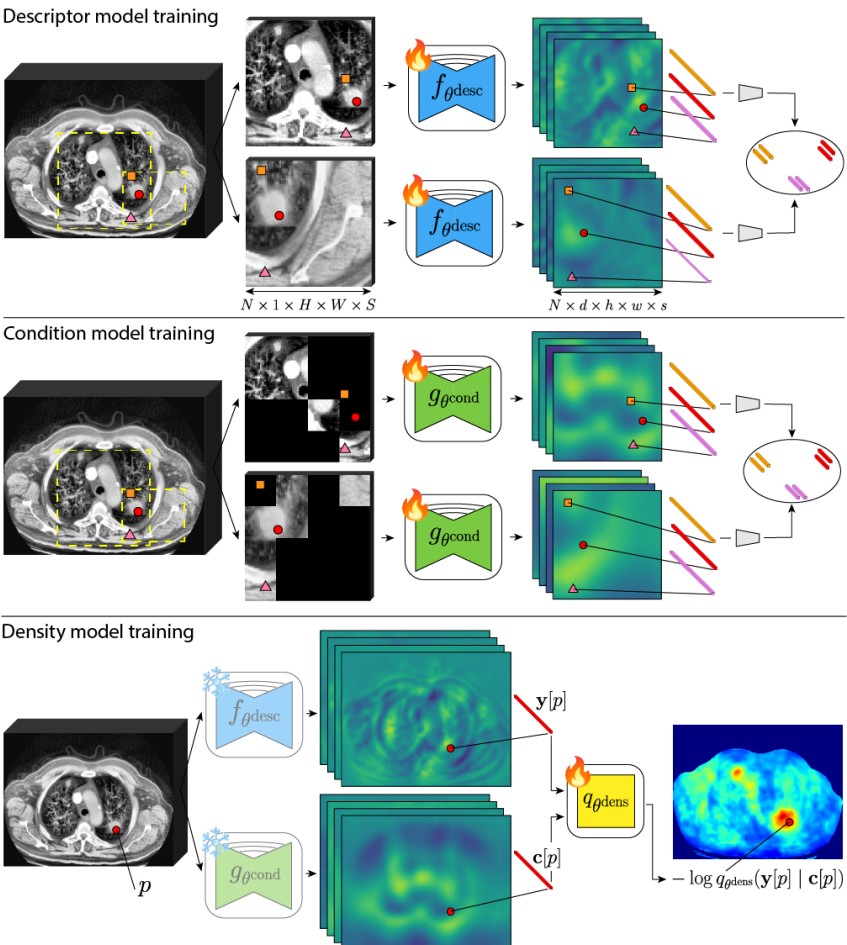

Figure 2: Illustration of Screener. First, we pretrain a *descriptor model* to produce discriminative feature maps, equivariant w.r.t. image crops and rescaling and invariant w.r.t. color jitter. Second, we train a *condition model* in the same way as the descriptor model, but also enforcing invariance to image masking. Third, a density model learns the conditional distribution $p_{Y|C}(y|c)$ of feature vectors $Y = y[p]$ and $C = c[p]$ extracted by the descriptor and condition models from random image at random position $p$. To obtain anomaly maps we apply the density model in a pixel-wise manner, which can be efficiently implemented using $1 \times 1 \times 1$ convolutions.

## 3.1 DESCRIPTOR MODEL

The success of our method relies on high-quality descriptors that are discriminative of pathology yet robust to irrelevant normal variations. Dense SSL provides a principled way to achieve this balance: voxel-level objectives encourage spatial discrimination, while augmentation invariance eliminates low-level details, leading to a smoother, more semantically structured embedding space in which similar normal patterns map to high-density areas.

Our descriptor model design follows domain-driven, minimalistic principles, differing from the prior dense SSL literature (Wang et al., 2021; Bardes et al., 2022). We adopt a UNet-like architecture, which has proven a strong dense feature extractor in 3D medical imaging. Full resolution output enables precise localization of small pathologies. Each training batch includes embeddings from nearby voxels, forcing distinction of even spatially adjacent locations. We omit auxiliary global objectives or multi-scale feature pyramids — our approach is simple and principled, relying solely on dense self-supervision at full resolution.

The training process is illustrated in the upper part of Figure 2. From a random CT volume $\mathbf{x}$, we extract two overlapping, randomly sized 3D crops, resize them to $H \times W \times S$, and apply augmentations such as color jitter. The augmented crops, denoted $\mathbf{x}^{(1)}$ and $\mathbf{x}^{(2)}$, are passed through the descriptor model to produce feature maps $\mathbf{y}^{(1)}$ and $\mathbf{y}^{(2)}$. From the overlapping region of the two crops, we randomly select $n$ positions. For each position $p$, we compute its coordinates $p^{(1)}$ and $p^{(2)}$ relative to the augmented views, and extract descriptors $y^{(1)} = \mathbf{y}^{(1)}[p^{(1)}]$ and $y^{(2)} = \mathbf{y}^{(2)}[p^{(2)}]$. These descriptors form a positive pair, as they correspond to the same position in the original volume but are predicted from different augmentations. Repeating this process for $m$ different seed CT volumes yields a batch of $N = n \cdot m$ positive pairs, denoted $\{(y_i^{(1)}, y_i^{(2)})\}_{i=1}^N$. These embeddings are then optimized using standard dense SSL objectives, such as InfoNCE (Chen et al., 2020) or VICReg (Bardes et al., 2021), described in Appendix B. We refer to the resulting models as *DenseInfoNCE* and *DenseVICReg*, respectively.

## 3.2 CONDITION MODEL

In medical imaging, the statistical plausibility of a local pattern often depends on its broader context, such as anatomical location or patient characteristics. This motivates modeling the conditional distribution of descriptors, given relevant contextual variables. Conditioning offers two key advantages: it simplifies density estimation, as conditional distributions are usually less complex than marginal, and it may lead to more semantically meaningful anomalies, defined as deviations from what is expected in a specific context. For example, a pattern normal in one anatomical region or patient group (e.g., a calcification in an elderly lung) might be abnormal in another (e.g., a calcification in breast).

Conditioning variables can be global (e.g., patient metadata like age or sex) or voxel-wise, enabling region-specific conditioning. General-domain UVAS methods (Gudovskiy et al., 2022; Zhou et al., 2024) utilize sinusoidal positional encodings of absolute spatial coordinates relative to the image origin. However, since medical scans may not be anatomically aligned, vanilla positional encodings lack consistent anatomical or patient-specific relevance. Anatomical Positional Embeddings (APE) (Goncharov et al., 2024) offer an alternative by encoding pixels' anatomical locations (though previously used for retrieval, not UVAS conditioning). However, it is domain-specific and may not capture all patient-level or fine-grained contextual nuances.

To address the aforementioned limitations, we propose a domain-agnostic self-supervised method for *learning* conditions. Our key idea is to train a *condition model* $g_{\theta^{\text{cond}}}$ to predict voxel-wise embeddings that are consistent across different masked image views. For instance, as illustrated in Figure 2, the model learns to predict the same condition embedding for a location even if a pathology is visible in one masked view but not another. Consequently, the learned condition feature maps become *invariant to the presence / absence of anomalies*. At the same time, we encourage intra-subject, i.e. spatial, and inter-subject discriminativeness and expect feature maps to capture voxel-level features such as anatomical location and tissue type, and patient-level characteristics such as age or sex, which are robustly inferable from the global image structure. We empirically probe pathology-related ignorance and anatomical awareness of our condition model in Appendix D. The architecture and training procedure for the condition model $g_{\theta^{\text{cond}}}$ are exactly the same as those for the descriptor model, with the sole difference: random masking as an additional augmentation.

## 3.3 DENSITY MODEL

The conditional density model $q_{\theta^{\text{dens}}}(y \mid c)$ can be viewed as a predictive model, which tries to predict descriptors based on the corresponding conditions. In this interpretation, anomaly scores $\{-\log q_{\theta^{\text{dens}}}(\mathbf{y}[p] \mid \mathbf{c}[p])\}_{p \in P}$ are position-wise prediction errors. Also note, that marginal density model $q_{\theta^{\text{dens}}}(y)$ is a special case of conditional model with a constant condition $\mathbf{c}[p] = \text{const}$.

During training, we sample a batch of $m$ random crops, $\{\mathbf{x}_i\}_{i=1}^m$, each of size $H \times W \times S$, from different CT images. For each crop, the pretrained descriptor and condition models produce the descriptor maps, $\{\mathbf{y}_i\}_{i=1}^m$, and condition maps, $\{\mathbf{c}_i\}_{i=1}^m$, and negative log-likelihood loss is optimized:

$$\min_{\theta_{\text{dens}}} \quad \frac{1}{m \cdot |P|} \sum_{i=1}^m \sum_{p \in P} -\log q_{\theta^{\text{dens}}}(\mathbf{y}_i[p] \mid \mathbf{c}_i[p]).$$

At inference, we divide an input CT image into $M$ overlapping patches, $\{\mathbf{x}_i\}_{i=1}^{M}$, each of size $H \times W \times S$. For each patch, we apply the descriptor, condition, and density models to compute the anomaly map, $\{-\log q_{\theta^{\mathrm{dens}}}(\mathbf{y}_i[p] \mid \mathbf{c}_i[p])\}_{p \in P}$. These patch-wise anomaly maps are then aggregated into a single anomaly map aligned with the entire input volume. During aggregation, we average the predictions in patches' overlapping regions.

We explore two parameterizations for the density model $q_{\theta^{\mathrm{dens}}}(y \mid c)$: Gaussian, as a straightforward baseline, and normalizing flows, similar to Gudovskiy et al. (2022); Zhou et al. (2024), as an expressive generative model enabling tractable density estimation. These parameterizations and the details of their implementation in the context of UVAS framework are further described in Appendix E.

### 3.4 Distillation and supervised fine-tuning

Although unsupervised Screener shows impressive results, supervised fine-tuning is the most practical way to further improve its performance. The density-based UVAS pipeline, consisting of three separate models, is not amenable to end-to-end optimization. To enable fine-tuning, we distill the knowledge from the pretrained Screener into a single UNet architecture. This step can be viewed as a novel self-supervised pretraining method for pathology segmentation tasks.

During distillation, we sample random image crops, pass them through the pretrained modular Screener to obtain ground truth anomaly score maps (negative log-density values). We then train a regression UNet model (last conv has one output channel without activation) to predict these score maps directly from the input image crops using a simple MSE loss. For supervised fine-tuning on binary segmentation tasks, we randomly reinitialize the UNet's last conv layer and append a sigmoid activation. Then we fine-tune the model on task-specific labeled data using a combination of voxel-wise binary cross-entropy and Dice losses.

## 4 Experiments

Our experiments can be divided into three main parts:

- **Unsupervised setting.** We show that our *unsupervised* Screener significantly outperforms other UVAS methods on real-word medical CT datasets (Section 4.1).

- **Fine-tuning setting.** We demonstrate that Screener can serve as a state-of-the-art self-supervised *pretraining* method. To this end, we fine-tune the distilled Screener (as described in Section 3.4) for pathology segmentation tasks and compare it with supervised model trained from scratch, as well as other fine-tuned pretrained models (Section 4.2).

- **Ablation study.** We explore how different choices of descriptor, condition and density models in our method affect the UVAS results (Section 4.3).

**Datasets.** We train Screener and other unsupervised models on three CT datasets: NLST (Team, 2011), AMOS (Ji et al., 2022), and AbdomenAtlas (Qu et al., 2024). These large-scale datasets include diverse patients with potential pathologies, but their annotations are not available for data filtering or training. For evaluation we use four datasets: LIDC (Armato III et al., 2011), MIDRC-RICORD-1a (Tsai et al., 2020), KiTS (Heller et al., 2019) and LiTS (Bilic et al., 2023). These datasets provide annotation masks only for certain pathologies. Any other pathologies present in these datasets are not labeled. Summary table about the datasets is provided in Appendix F.

### 4.1 Unsupervised setting

**Evaluation protocol.** We compare Screener with baseline UVAS models using voxel-level AUROC and Dice score. The segmentation threshold is selected to maximize Dice score on a subset of 10 pathological cases, and then kept fixed for the entire test set. Note that Dice scores are significantly underestimated due to incomplete ground truth masks: while UVAS models aim to detect *all* anomalies, the datasets provide annotations only for specific target pathologies. Detections corresponding to other unlabeled pathologies (see Figure 1 and Appendix A) are therefore mistakenly counted as false positives. Voxel-level AUROC is a standard UVAS metric because its estimation is more robust to the ground truth incompleteness issue. We estimate AUROC across all dataset

voxels by sampling 1000 pathological voxels (contributing to true positive rate) and 1000 out-of-mask "normal" voxels (for false positive rate) per test image. The sampled "normal" voxels are overwhelmingly normal, ensuring accurate AUROC estimation despite incomplete annotations.

**Results.** Quantitative results are presented in Table 1. Qualitative results are shown in Figure 3. Screener significantly outperforms the UVAS baselines. Autoencoder, f-AnoGAN and Patched Diffusion Model tend to overfit to pathologies in the training data, and fail to reconstruct fine-grained normal details (see also Appendix I). Synthetics-based DRAEM and MOOD-Top1 struggle to generalize to the appearance of real medical pathologies. The density-based MSFlow, relying on ImageNet-pretrained features, proves ineffective at discriminating pathologies from normal regions in CT images.

Table 1: Comparison of Screener and the existing UVAS methods in unsupervised setting.

| Model | Voxel-level AUROC | | | | Dice score[1] | | | |
|---|---|---|---|---|---|---|---|---|
| | LIDC | MIDRC | KiTS | LiTS | LIDC | MIDRC | KiTS | LiTS |
| Autoencoder (Baur et al., 2021) | 0.71 | 0.65 | 0.66 | 0.68 | $0.00 \pm 0.00$ | $0.09 \pm 0.07$ | $0.01 \pm 0.02$ | $0.01 \pm 0.01$ |
| f-AnoGAN (Schlegl et al., 2019) | 0.82 | 0.66 | 0.67 | 0.67 | $0.00 \pm 0.00$ | $0.09 \pm 0.07$ | $0.01 \pm 0.02$ | $0.01 \pm 0.01$ |
| Patched Diffusion Model (Behrendt et al., 2024) | 0.87 | 0.76 | 0.76 | 0.80 | $0.01 \pm 0.03$ | $0.14 \pm 0.08$ | $0.02 \pm 0.03$ | $0.02 \pm 0.04$ |
| DRAEM (Zavrtanik et al., 2021) | 0.63 | 0.72 | 0.82 | 0.83 | $0.00 \pm 0.00$ | $0.11 \pm 0.08$ | $0.03 \pm 0.06$ | $0.02 \pm 0.04$ |
| MOOD-Top1 (Marimont & Tarroni, 2023) | 0.79 | 0.79 | 0.77 | 0.80 | $0.00 \pm 0.01$ | $0.13 \pm 0.10$ | $0.02 \pm 0.07$ | $0.06 \pm 0.12$ |
| MSFlow (Zhou et al., 2024) | 0.71 | 0.67 | 0.63 | 0.63 | $0.00 \pm 0.01$ | $0.08 \pm 0.06$ | $0.01 \pm 0.01$ | $0.00 \pm 0.01$ |
| Screener (ours) | **0.96** | **0.87** | **0.90** | **0.93** | **$0.05 \pm 0.13$** | **$0.30 \pm 0.18$** | **$0.06 \pm 0.09$** | **$0.10 \pm 0.12$** |

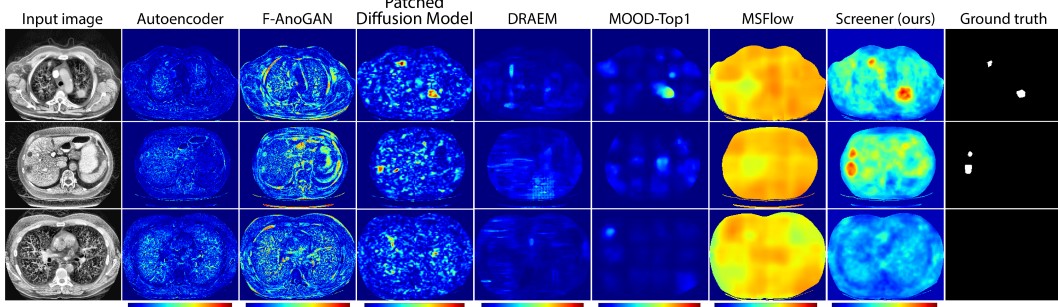

Figure 3: Qualitative comparison of anomaly maps produced by baseline UVAS methods and unsupervised Screener. First column contains CT slices, columns 2 to 7 are the baseline methods' predictions, column 8 is the Screener's prediction. Last column depicts the ground truth mask.

## 4.2 FINE-TUNING SETTING

**Evaluation protocol.** We fine-tune and test pretrained models on the evaluation datasets via 3-fold cross-validation. For each training fold, we use only 25 labeled cases, to amplify pretraining benefits and to conserve computational resources. Full-data fine-tuning results can be found in Appendix J. We assess the models using Dice score. We use a Wilcoxon signed-rank test to compare all the fine-tuned models with the nnUNet (Isensee et al., 2021) trained from scratch.

**Results.** Results in Table 2 demonstrate that Screener-based pretraining consistently improves downstream segmentation performance across all test datasets in low-data fine-tuning setup, with significant gains on LIDC (a 1.5-fold Dice increase) and LiTS. Screener is competitive with supervised pretraining (Huang et al., 2023) and state-of-the-art self-supervised VoCo (Wu et al., 2024), and outperform other SSL models (Zhou et al., 2021; Tang et al., 2022; Valanarasu et al., 2023).

---

[1]Note that Dice scores are often underestimated in the unsupervised setting, as ground truth masks cover only certain target pathologies, while UVAS models intentionally detect *all* pathologies. Many true positives are thus mistakenly counted as false positives (see Figure 1 and Appendix A for examples).

However, when fine-tuned on full datasets, Screener does not yield any gains compared to training from scratch (see Appendix J), which is consistent with the broader observation that many SSL methods provide their strongest advantages when training data are scarce.

Table 2: Dice scores of Screener and other self-supervised pretrained models after fine-tuning. We highlight statistically significant improvements (green) or declines (red) relative to nnUNet trained from random initialization. Note that Grounding DINO (Liu et al., 2023) and LVM-Med (MH Nguyen et al., 2023) operate on 2D slices without access to full 3D image context, and therefore show inferior performance to other baselines which use 3D convolutions or 3D attention.

| Model | LIDC | MIDRC | KiTS | LiTS |
|---|---|---|---|---|
| nnUNet (random init.) (Isensee et al., 2021) | 0.21 | **0.61** | 0.41 | 0.45 |
| nnUNet (supervised pretrain.) (Huang et al., 2023) | **0.29** ↑ 40% ($p < 0.01$) | **0.62** ↑ 2% ($p = 0.51$) | 0.46 ↑ 10% ($p < 0.01$) | **0.48** ↑ 7% ($p < 0.01$) |
| Grounding DINO (Liu et al., 2023) | 0.09 ↓ 54% ($p < 0.01$) | 0.58 ↓ 5% ($p < 0.01$) | 0.19 ↓ 55% ($p < 0.01$) | 0.33 ↓ 26% ($p < 0.01$) |
| LVM-Med (MH Nguyen et al., 2023) | 0.04 ↓ 80% ($p < 0.01$) | 0.55 ↓ 8% ($p < 0.01$) | 0.16 ↓ 62% ($p < 0.01$) | 0.33 ↓ 26% ($p < 0.01$) |
| Model Genesis (Zhou et al., 2021) | 0.21 ↑ 1% ($p = 0.76$) | 0.59 ↓ 2% ($p = 0.05$) | 0.34 ↓ 18% ($p < 0.01$) | 0.39 ↓ 12% ($p = 0.01$) |
| SwinUNETR (Tang et al., 2022) | 0.16 ↓ 24% ($p < 0.01$) | 0.55 ↓ 9% ($p < 0.01$) | 0.19 ↓ 53% ($p < 0.01$) | 0.39 ↓ 13% ($p < 0.01$) |
| DAE (Valanarasu et al., 2023) | 0.15 ↓ 26% ($p < 0.01$) | 0.58 ↓ 4% ($p < 0.01$) | 0.26 ↓ 38% ($p < 0.01$) | 0.36 ↓ 20% ($p < 0.01$) |
| VoCo (Wu et al., 2024) | 0.20 ↓ 2% ($p = 0.79$) | **0.61** ↑ 1% ($p = 0.89$) | 0.49 ↑ 17% ($p < 0.01$) | 0.49 ↑ 10% ($p < 0.01$) |
| MAE (Wald et al., 2025) | 0.24 ↑ 18% ($p < 0.01$) | **0.62** ↑ 3% ($p = 0.01$) | 0.35 ↓ 17% ($p = 0.04$) | 0.49 ↑ 10% ($p < 0.01$) |
| DenseVICReg | 0.22 ↑ 7% ($p = 0.15$) | 0.58 ↓ 4% ($p < 0.01$) | 0.31 ↓ 26% ($p < 0.01$) | 0.44 ↓ 2% ($p = 0.92$) |
| Screener (ours) | **0.31** ↑ 49% ($p < 0.01$) | **0.62** ↑ 3% ($p = 0.45$) | 0.43 ↑ 4% ($p = 0.17$) | **0.48** ↑ 7% ($p < 0.01$) |

## 4.3 ABLATION STUDY

Table 3 presents the ablation study of our proposed condition model. We compare our condition model with two baselines: vanilla sinusoidal positional encodings and APE (Goncharov et al., 2024), detailed in Appendix C. We evaluate condition models in combination with the fixed DenseVICReg descriptor model and two different density models — Gaussian and normalizing flow — described in Appendix E. When we use expressive normalizing flow density model, all conditioning strategies yield results comparable to each other and to the unconditional model. However, in experiments with simple Gaussian density models, we see that the results significantly improve as the conditioning variables becomes more informative. Remarkably, our proposed masking-invariant condition model allows Gaussian model to achieve very strong anomaly segmentation results competing with complex flow-based models.

Table 3: Ablation study of the effect of conditional model for gaussian and flow-based density models. None in Condition model column means that results are given for a marginal model.

| Descriptor model | Condition model | Density model | Voxel-level AUROC | | | | Dice score | | | |
|---|---|---|---|---|---|---|---|---|---|---|
| | | | LIDC | MIDRC | KiTS | LiTS | LIDC | MIDRC | KiTS | LiTS |
| DenseVICReg, $d^{\text{desc}} = 32$ | None | Gaussian | 0.81 | 0.81 | 0.61 | 0.71 | $0.00 \pm 0.00$ | $0.17 \pm 0.13$ | $0.00 \pm 0.01$ | $0.00 \pm 0.01$ |
| —— " —— | Sin-cos pos. | —— " —— | 0.82 | 0.80 | 0.74 | 0.77 | $0.00 \pm 0.00$ | $0.14 \pm 0.11$ | $0.01 \pm 0.02$ | $0.01 \pm 0.02$ |
| —— " —— | APE | —— " —— | 0.88 | 0.80 | 0.78 | 0.86 | $0.00 \pm 0.03$ | $0.14 \pm 0.10$ | $0.01 \pm 0.01$ | $0.01 \pm 0.03$ |
| —— " —— | Masking-invariant | —— " —— | **0.96** | **0.84** | **0.87** | **0.90** | $\mathbf{0.04 \pm 0.08}$ | $\mathbf{0.21 \pm 0.13}$ | $\mathbf{0.03 \pm 0.05}$ | $\mathbf{0.13 \pm 0.19}$ |
| —— " —— | None | Norm. flow | **0.96** | **0.89** | 0.88 | 0.93 | $\mathbf{0.05 \pm 0.12}$ | $\mathbf{0.31 \pm 0.18}$ | $0.04 \pm 0.06$ | $0.09 \pm 0.12$ |
| —— " —— | Sin-cos pos. | —— " —— | **0.96** | **0.89** | **0.90** | **0.94** | $\mathbf{0.05 \pm 0.13}$ | $0.30 \pm 0.18$ | $0.06 \pm 0.09$ | $\mathbf{0.10 \pm 0.12}$ |
| —— " —— | APE | —— " —— | **0.96** | 0.88 | 0.89 | **0.94** | $0.04 \pm 0.11$ | $0.28 \pm 0.18$ | $0.05 \pm 0.08$ | $0.09 \pm 0.13$ |
| —— " —— | Masking-invariant | —— " —— | **0.96** | 0.87 | **0.90** | 0.93 | $\mathbf{0.05 \pm 0.13}$ | $0.28 \pm 0.18$ | $\mathbf{0.07 \pm 0.11}$ | $0.10 \pm 0.13$ |

We also ablate different choices of descriptor model in Table 4. We compare DenseInfoNCE and DenseVICReg and conclude that dense VICReg objective works slightly better. We also compare two DenseVICReg models with different descriptors' dimensionality $d^{\text{desc}} = 32$ or $d^{\text{desc}} = 128$ and conclude that increasing dimensionality does not improve the results. To demonstrate the superiority of our domain-specific self-supervised descriptor model over supervised feature extractors, we compare them it with ImageNet-pretrained ResNet50 (Zhou et al., 2024) and STU-Net (Huang et al., 2023) — a UNet pretrained in a supervised manner on anatomical structure segmentation tasks.

## 5 RELATED WORK

**Reconstruction-based UVAS.** Reconstruction-based methods train a generative model to reconstruct the original image from its compressed representation (Baur et al., 2021; Schlegl et al., 2019)

Table 4: Ablation study of the effect of descriptor model. In these experiments we do not use conditioning and use normalizing flow as a marginal density model. We include MSFlow (Zhou et al., 2024) to demonstrate that ImageNet-pretrained descriptor model is inappropriate for 3D medical CT images.

| Descriptor model | Condition model | Density model | Voxel-level AUROC | | | | Dice score | | | |
|---|---|---|---|---|---|---|---|---|---|---|
| | | | LIDC | MIDRC | KiTS | LiTS | LIDC | MIDRC | KiTS | LiTS |
| ImageNet | Sin-cos pos. | MSFlow | 0.70 | 0.66 | 0.64 | 0.64 | $0.00 \pm 0.01$ | $0.08 \pm 0.06$ | $0.01 \pm 0.01$ | $0.00 \pm 0.01$ |
| STU-Net (Huang et al., 2023) | None | Norm. flow | 0.52 | 0.44 | 0.52 | 0.64 | $0.00 \pm 0.00$ | $0.02 \pm 0.03$ | $0.01 \pm 0.02$ | $0.01 \pm 0.01$ |
| DenseInfoNCE, $d^{\text{desc}} = 32$ | None | Norm. flow | **0.96** | 0.87 | 0.87 | 0.91 | $0.04 \pm 0.11$ | $0.28 \pm 0.18$ | **0.04 ± 0.06** | $0.05 \pm 0.09$ |
| DenseVICReg, $d^{\text{desc}} = 32$ | None | Norm. flow | **0.96** | 0.89 | **0.88** | **0.93** | **0.05 ± 0.12** | **0.31 ± 0.18** | **0.04 ± 0.06** | **0.09 ± 0.12** |
| DenseVICReg, $d^{\text{desc}} = 128$ | None | Norm. flow | **0.96** | **0.90** | 0.87 | **0.93** | $0.04 \pm 0.09$ | **0.31 ± 0.18** | $0.03 \pm 0.06$ | $0.08 \pm 0.12$ |

or from its corrupted, e.g., noised (Behrendt et al., 2024), version. If training set is anomaly-free these models struggle to reconstruct anomalies in the test set and absolute differences between the original and reconstructed pixel values can be used as anomaly maps. However, when training dataset contains real anomalies, reconstruction-based models can learn to reconstruct anomalies as well as normal regions, diminishing their ability to differentiate. Another limitation is that measuring reconstruction errors in raw pixel space can be problematic: some abnormal pixels can accidentaly have small reconstruction errors, while some normal fine-grained details, which are inherently difficult to reconstruct precisely, might yield high reconstruction errors.

**Synthetics-based UVAS.** These methods rely on generating synthetic image anomalies and training a supervised model to segment them. Anomalies can be simulated by corrupting random image regions with noise, replacing them with random patterns from a specialized set (Zavrtanik et al., 2021), or using parts of other training images (Marimont & Tarroni, 2023). While these models are straightforward to implement and train, they overfit to synthetic anomalies and struggle to generalize effectively to real-world anomalies.

**Density-based UVAS.** We explain the idea of density-based UVAS in Section 2.1. Some methods (Roth et al., 2022) use non-parametric density models based on memory banks. More scalable flow-based methods (Yu et al., 2021; Gudovskiy et al., 2022; Zhou et al., 2024), leverage normalizing flows. In our experiments, we included MSFlow (Zhou et al., 2024), as it was among the top-5 performing methods on MVTecAD (Bergmann et al., 2021) at the time.

**Medical UVAS.** Recognized methods are either reconstruction-based (Baur et al., 2021; Schlegl et al., 2019; Pinaya et al., 2022; Behrendt et al., 2024) or synthetics-based (Marimont & Tarroni, 2023). f-AnoGAN (Schlegl et al., 2019) trains generator $g$ and discriminator $d$, to generate anomaly-free images $x \sim g(z)$ from latent variables $z$. Then, it trains encoder $f$ to map anomaly-free images $x$ to the latent space, s.t. they can be reconstructed via frozen generator $\hat{x} = g(f(x)) \approx x$. Patched Diffusion Model (Behrendt et al., 2024) cuts out image patches and trains a diffusion model to reconstruct them based on the surrounding context. At inference, an image is split into a grid of patches and Diffusion model reconstructs each patch from its noised version based on the remaining clean patches. MOOD-Top1 (Marimont & Tarroni, 2023) is a straightforward synthetics-based method showing top-1 performance on MOOD (Zimmerer et al., 2022).

**Medical self-supervised pretraining.** Methods like Model Genesis (Zhou et al., 2021) and SwinUNETR (Tang et al., 2022) utilize combinations of contrastive learning, masked image modeling, and various pretext tasks re-implemented for 3D CT volumes. DAE (Valanarasu et al., 2023) pretrain a model to reconstruct original images from their disrupted versions created by local masking across channel embeddings and low-level perturbations like noise and downsampling. Volume Contrast (VoCo) (Wu et al., 2024) employs a contrastive approach to implicitly encode contextual position priors, treating different image regions as distinct "classes" and predicting which region a random sub-volume belongs to by contrasting its representation against base crops. To our knowledge, Screener is the first work to propose and demonstrate the effectiveness of using unsupervised anomaly segmentation as a pretraining strategy for downstream pathology segmentation tasks.

# 6 CONCLUSION

Our work addresses the critical challenge of detecting all pathological findings in 3D CT images, a task hindered by limited labeled data. Assuming the inherent rarity of pathological patterns, we frame this as a UVAS problem. We propose Screener, a novel density-based UVAS framework with dense SSL, ensuring discriminative and robust domain-specific descriptors, and learned, masking-invariant conditioning variables that simplify density modeling. Evaluated on four large-scale datasets, the fully unsupervised Screener achieved state-of-the-art performance, effectively localizing diverse pathologies. Furthermore, when distilled and fine-tuned, Screener demonstrated strong performance on supervised segmentation tasks, establishing its value as a novel pretraining method. Screener represents a significant step towards comprehensive and scalable pathology detection, serving as a powerful unsupervised screening tool and a robust foundation for supervised applications.

**Limitations & future work.** Despite its promising performance, Screener has several limitations that warrant future investigation. Its reliance on the rarity assumption may lead to false negative errors for common or widespread pathologies, while statistical anomalies that lack clinical significance, e.g. artifacts, could result in false positives (though we analyze robustness to artifacts, low-dose and contrast agent in Appendix H). Comprehensive evaluation of UVAS methods remains challenging due to the lack of ground truth annotations for all potential pathologies. Currently validated on CT, the generalizability of our approach to other medical imaging modalities requires further exploration. Future work will also explore scaling laws to investigate how performance scales with model size and training data, potentially unlocking further improvements.

## ACKNOWLEDGMENTS

This work was supported by The Ministry of Economic Development of the Russian Federation in accordance with the subsidy agreement (agreement identifier 000000C313925P4H0002; grant No 139-15-2025-012).

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

## A   DICE SCORES UNDERESTIMATION IN UNSUPERVISED SETTING

In Figure 4 we give examples of reasonable Screener's predictions for which Dice scores cannot be calculated properly due to incomplete ground truth masks.

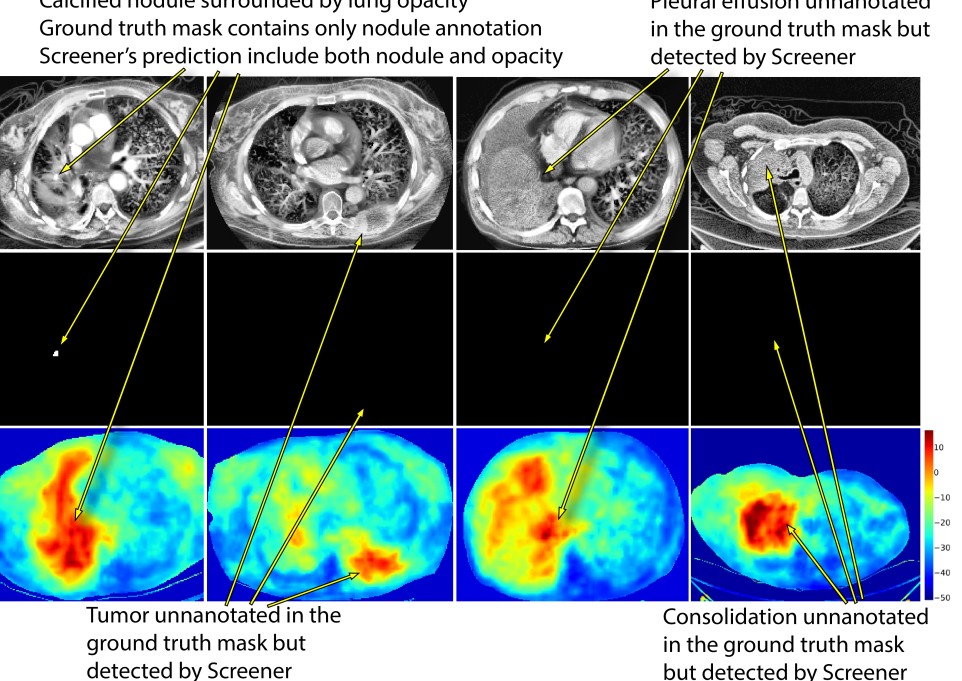

Figure 4: Examples of Screener's **true positive predictions** (third row) counted as "false positives" due to incompleteness of the ground truth masks (second row), leading to Dice score underestimation.

## B   SELF-SUPERVISED LEARNING

**InfoNCE.**   As in SimCLR (Chen et al., 2020), batch of positive pairs $\{(y_i^{(1)}, y_i^{(2)})\}_{i=1}^N$ is passed through a trainable MLP-projector $g_{\theta\text{proj}}$ and L2-normalized: $z_i^{(k)} = g_{\theta\text{proj}}(y_i^{(k)})/\|g_{\theta\text{proj}}(y_i^{(k)})\| \in \mathbb{R}^d$, where $k = 1, 2$ and $i = 1, \dots, N$. Then, the objective is to maximize similarity in positive pairs while minimizing similarity in negative pairs. To this end, InfoNCE loss is written as:

$$\min_\theta \quad \sum_{i=1}^N \sum_{k\in\{1,2\}} -\log \frac{\exp(\langle z_i^{(1)}, z_i^{(2)}\rangle/\tau)}{\exp(\langle z_i^{(1)}, z_i^{(2)}\rangle/\tau) + \sum_{j\neq i}\sum_{l\in\{1,2\}}\exp(\langle z_i^{(k)}, z_j^{(l)}\rangle/\tau)}. \tag{1}$$

**VICReg.**   VICReg objective consists of three terms:

$$\min_\theta \quad \alpha \cdot \mathcal{L}^{\text{inv}} + \beta \cdot \mathcal{L}^{\text{var}} + \gamma \cdot \mathcal{L}^{\text{cov}}. \tag{2}$$

The first term enforces embeddings to be invariant to augmentations:

$$\mathcal{L}^{\text{inv}} = \frac{1}{N \cdot D}\sum_{i=1}^N \|z_i^{(1)} - z_i^{(2)}\|^2. \tag{3}$$

The second term ensures that individual embeddings' dimensions have a least unit variance:

$$\mathcal{L}^{\text{var}} = \sum_{k\in\{1,2\}} \frac{1}{D}\sum_{i=1}^D \max\left(0, 1 - \sqrt{C_{i,i}^{(k)} + \varepsilon}\right). \tag{4}$$

The third term encourages different embeddings' dimensions to be uncorrelated, increasing the total information content of the embeddings:

$$\mathcal{L}^{\mathrm{cov}} = \sum_{k \in \{1,2\}} \frac{1}{D} \sum_{i \neq j} \left( C_{i,j}^{(k)} \right)^2 . \tag{5}$$

In VICReg, embeddings $\{z_i^{(k)}\}$ are not L2-normalized and obtained through a trainable MLP-expander which increases the dimensionality up to 8192.

## C  BASELINE CONDITION MODELS

**Sin-cos positional encodings.**   The existing density-based UVAS methods (Gudovskiy et al., 2022; Zhou et al., 2024) for natural images use standard sin-cos positional encodings for conditioning. We also employ them as an option for condition model in our framework. However, let us clarify what we mean by sin-cos positional embeddings in CT images. Note that we never apply descriptor, condition or density models to the whole CT images due to memory constraints. Instead, at all the training stages and at the inference stage of our framework we always apply them to image crops of size $H \times W \times S$, as described in Sections 3.1 and 3.3. When we say that we apply sin-cos positional embeddings condition model to an image crop, we mean that compute sin-cos encodings of absolute positions of its pixels w.r.t. to the whole CT image.

**Anatomical positional embeddings.**   To implement the idea of learning the conditional distribution of image patterns at each certain anatomical region, we need a condition model producing conditions $c[p]$ that encode which anatomical region is present in the image at every position $p$. Supervised model for organs' semantic segmentation would be an ideal condition model for this purpose. However, to our best knowledge, there is no supervised models that are able to segment all organs in CT images. That is why, we decided to try the self-supervised APE (Goncharov et al., 2024) model which produces continuous embeddings of anatomical position of CT image pixels.

## D  DESCRIPTOR AND CONDITION MODELS PROBING ANALYSIS

Our intuition — introduced in Section 3, and empirically validated in Table 3 — is that an effective conditioning variables should exclude information about presence / absence of pathologies, while capturing as much contextual information as possible (e.g., anatomical region, tissue type, patient-level characteristics etc.). This combination simplifies conditional density modeling and sharpens anomaly detection.

To make this intuition explicit, we conducted an additional probing experiment comparing the amount of pathology-related information retained by the descriptor model versus the condition model. We trained a lightweight segmentation head on top of frozen descriptor vs. condition embeddings and evaluated Dice scores on four pathology segmentation tasks. As seen in Table 5, our condition model probing yields substantially lower Dice scores (p-value $< 10^{-10}$ for all datasets). For lung cancer and liver tumors, scores are essentially zero, confirming that pathology-specific information is effectively removed. For pneumonia and kidney tumors, non-zero Dice arises largely because these pathologies occupy substantial portions of the corresponding organs; the segmentation head tends to predict the whole organ, producing non-zero Dice scores.

Table 5: Probing of frozen descriptor model (DenseVICReg) and condition model (masking-invariant DenseVICReg) feature maps on supervised pathology segmentation tasks (results are Dice scores).

| Model | LIDC | MIDRC | KiTS | LiTS |
|---|---|---|---|---|
| Descriptor model probe | $\mathbf{0.19 \pm 0.23}$ | $\mathbf{0.40 \pm 0.19}$ | $\mathbf{0.17 \pm 0.18}$ | $\mathbf{0.25 \pm 0.23}$ |
| Condition model probe | $0.00 \pm 0.00$ | $0.27 \pm 0.16$ | $0.06 \pm 0.10$ | $0.01 \pm 0.03$ |

To demonstrate that our condition model indeed captures anatomical information better than base-lines, we also probe it on *liver* segmentation task on LiTS. As seen in Table 6, condition model

embeddings separate liver vs. non-liver voxels substantially better than APE and sinusoidal positional encodings.

Table 6: Probing of our masking-invariant condition model vs. APE vs. sin-cos positional embeddings on supervised liver segmentation task (results are Dice scores).

| Model | LiTS (liver) |
|---|---|
| Condition model probe | $0.82 \pm 0.14$ |
| APE probe | $0.62 \pm 0.13$ |
| Sin-cos pos. embed. probe | $0.36 \pm 0.14$ |

# E   DENSITY MODELS

Below, we describe simple Gaussian density model and more expressive learnable Normalizing Flow model.

**Gaussian** marginal density model is written as

$$- \log q_{\theta^{\text{dens}}}(y) = \frac{1}{2}(y - \mu)^\top \Sigma^{-1}(y - \mu) + \frac{1}{2} \log \det \Sigma + \text{const}, \tag{6}$$

where the trainable parameters $\theta^{\text{dens}}$ are mean vector $\mu$ and diagonal covariance matrix $\Sigma$.

Conditional Gaussian density model is written as

$$- \log q_{\theta^{\text{dens}}}(y \mid c) = \frac{1}{2}(y - \mu_{\theta^{\text{dens}}}(c))^\top \left(\Sigma_{\theta^{\text{dens}}}(c)\right)^{-1} (y - \mu_{\theta^{\text{dens}}}(c)) + \frac{1}{2} \log \det \Sigma_{\theta^{\text{dens}}}(c) + \text{const}, \tag{7}$$

where $\mu_{\theta^{\text{dens}}}$ and $\Sigma_{\theta^{\text{dens}}}$ are MLP nets which take condition $c \in \mathbb{R}^{d^{\text{cond}}}$ as input and predict a conditional mean vector $\mu_{\theta^{\text{dens}}}(c) \in \mathbb{R}^{d^{\text{desc}}}$ and a vector of conditional variances which is used to construct the diagonal covariance matrix $\Sigma_{\theta^{\text{dens}}}(c) \in \mathbb{R}^{d^{\text{desc}} \times d^{\text{desc}}}$.

As described in Section 3.3, at both training and inference stages, we need to obtain dense negative log-density maps. Dense prediction by MLP nets $\mu_{\theta^{\text{dens}}}(c)$ and $\Sigma_{\theta^{\text{dens}}}(c)$ can be implemented using convolutional layers with kernel size $1 \times 1 \times 1$. In practice, we increase this kernel size to $3 \times 3 \times 3$, which can be equivalently formulated as conditioning on locally aggregated conditions.

**Normalizing flow** model of descriptors' marginal distribution is written as:

$$- \log p_{\theta^{\text{dens}}}(y) = \frac{1}{2} \| f_{\theta^{\text{dens}}}(y) \|^2 - \log \left| \det \frac{\partial f_{\theta^{\text{dens}}}(y)}{\partial y} \right| + \text{const}, \tag{8}$$

where neural net $f_\theta$ must be invertible and has a tractable Jacobian determinant.

Conditional normalizing flow model of descriptors' conditional distribution is given by:

$$- \log p_{\theta^{\text{dens}}}(y \mid c) = \frac{1}{2} \| f_{\theta^{\text{dens}}}(y, c) \|^2 - \log \left| \det \frac{\partial f_{\theta^{\text{dens}}}(y, c)}{\partial y} \right| + \text{const}, \tag{9}$$

where neural net $f_\theta: \mathbb{R}^{d^{\text{desc}}} \times \mathbb{R}^{d^{\text{cond}}} \to \mathbb{R}^{d^{\text{desc}}}$ must be invertible w.r.t. the first argument, and the second term should be tractable.

We construct $f_\theta$ by stacking Glow layers (Kingma & Dhariwal, 2018): act-norms, invertible linear transforms and affine coupling layers. Note that at both training and inference stages we apply $f_\theta$ to descriptor maps $\mathbf{y} \in \mathbb{R}^{h \times w \times s \times d^{\text{desc}}}$ in a pixel-wise manner to obtain dense negative log-density maps. In conditional model, we apply conditioning in affine coupling layers similar to Gudovskiy et al. (2022) and also in each act-norm layer by predicting maps of rescaling parameters based on condition maps.

## F    DATASETS

We utilized several publicly available datasets for training and evaluation summarized in Table 7. For training, we used the NLST (Team, 2011), AMOS (Ji et al., 2022), and AbdomenAtlas (Qu et al., 2024) datasets. NLST data access is controlled by the National Cancer Institute Data Access Committee and is available for research use. AMOS is released under a Creative Commons Attribution-NonCommercial-ShareAlike 4.0 International License (CC BY-NC-SA 4.0). AbdomenAtlas is licensed under CC BY-NC-SA 4.0 and intended for academic, research, and educational purposes. For evaluation, we used the LIDC-IDRI (LIDC) (Armato III et al., 2011), MIDRC-RICORD-1a (MIDRC) (Tsai et al., 2020), KiTS (Heller et al., 2019), and LiTS (Bilic et al., 2023) datasets. LIDC-IDRI is available through The Cancer Imaging Archive (TCIA) and is typically used under terms permitting research and education. MIDRC-RICORD-1a is also available through TCIA under similar terms, permitting non-commercial use for research and education. The KiTS dataset (version 2021) is available under a CC BY-NC-SA 4.0 license, primarily for non-commercial research and educational purposes. The LiTS dataset is available for research purposes, often under a Creative Commons Attribution-NonCommercial-NoDerivatives 4.0 International License (CC BY-NC-ND 4.0) or similar terms, as specified by its organizers. We have used all datasets in accordance with their specified licenses and terms of use.

Table 7: Summary information on the datasets that we use for training and testing of all models.

| Dataset | # 3D images | Annotated pathology |
|---|---|---|
| NLST (Team, 2011) | 25,652 | – |
| AMOS (Ji et al., 2022) | 2,123 | – |
| AbdomenAtlas (Qu et al., 2024) | 4,607 | – |
| LIDC (Armato III et al., 2011) | 1,017 | lung cancer |
| MIDRC (Tsai et al., 2020) | 115 | pneumonia |
| KiTS (Heller et al., 2019) | 298 | kidney tumors |
| LiTS (Bilic et al., 2023) | 117 | liver tumors |

## G    IMPLEMENTATION DETAILS

We preprocess CT volumes by cropping them to dense foreground voxels (thresholded by $-500$HU), resizing to $1.5 \times 1.5 \times 2.25$ mm$^3$ voxel spacing, clipping intensities to $[-1000, 300]$HU, rescaling them to $[0, 1]$ range and applying CLAHE (Pizer et al., 1987) which ensures that color jitter augmentations preserve information about presence of pathologies during descriptor model training.

We train both the descriptor model and the condition model for 300k batches of $m = 8$ pairs of overlapping patches with $N = 8192$ positive pairs of voxels. The training takes about 3 days on a single NVIDIA RTX H100-80GB GPU. We use AdamW optimizer, warm-up learning rate from 0.0 to 0.0003 during first 10K batches, and then reduce it to zero till the end of the training. Weight decay is set to $10^{-6}$ and gradient clipping to 1.0 norm. Patch size is set to $H \times W \times S = 96 \times 96 \times 64$.

During density model training, we apply average pooling operations with the $3 \times 3 \times 2$ stride to feature maps produced by the descriptor model and the condition model, following (Gudovskiy et al., 2022; Zhou et al., 2024). Thus $h \times w \times s = 32 \times 32 \times 32$. We inject Gaussian noise with 0.1 standard deviation both to the descriptors and conditions in order to stabilize the training. We train the density model for 500k batches each containing $m = 4$ patches. This training stage again takes about 3 days on a single NVIDIA RTX H100-80GB GPU. We use the same optimizer and the learning rate scheduler as for the descriptor and condition models.

The modular Screener model has 133M parameters, patch-based inference for a whole CT volume on NVIDIA RTX H100 GPU requires 4 Gb of GPU memory and takes about 5-10 seconds depending on the number of slices. The distilled Screener has 350M parameters, its patch-based inference requires 5 Gb of GPU memory and takes 0.5-1.0 seconds. We did not observe any difference in quality metrics for the distilled model compared to the modular model.

## H    ROBUSTNESS ANALYSIS

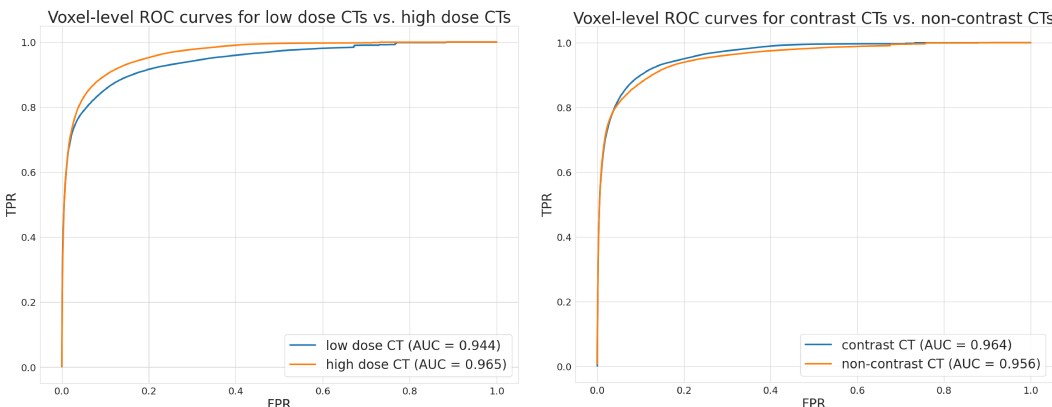

Figure 5: Comparison of Screener's voxel-level AUROCs on high-dose vs. low-dose and on contrast vs. non-contrast images from LIDC dataset.

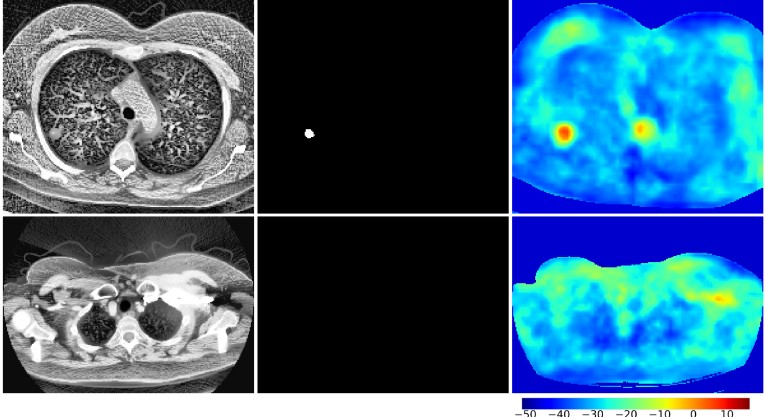

Figure 6: Examples of Screener performance on low-dose CT and artifacts. First row: Screener successfully segments lung cancer in low-dose CT. Second row: Screener assigns high anomaly scores to artifact.

# I    ANALYSIS OF RECONSTRUCTION-BASED MODELS

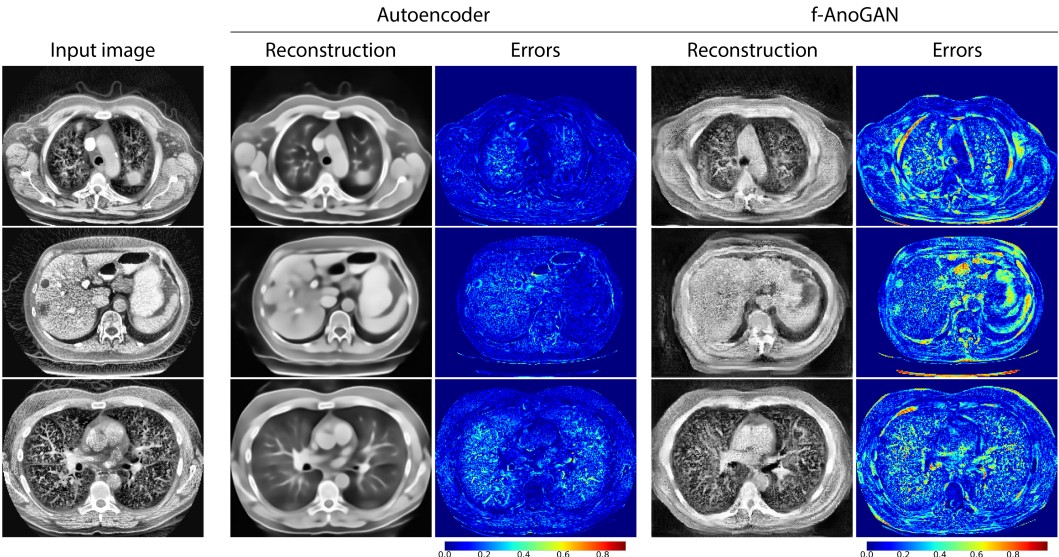

Figure 7: Reconstructions and anomaly maps predicted by Autoencoder Baur et al. (2021) (second and third columns) and f-AnoGAN Schlegl et al. (2019) (last two columns). Autoencoder overfits to reconstruct pathologies and thus fails to detect them. Also Autoencoder produces blurry generations, leading to inaccurate reconstructions and high anomaly scores on fine details (e.g., vessels in the lungs). f-AnoGAN avoids generating pathologies, but the reconstruction quality still is insufficient, resuling in false positive errors. GANs are known to be unstable and sensitive to hyperparameters, necessitating careful tuning and experimentation to achieve optimal results.

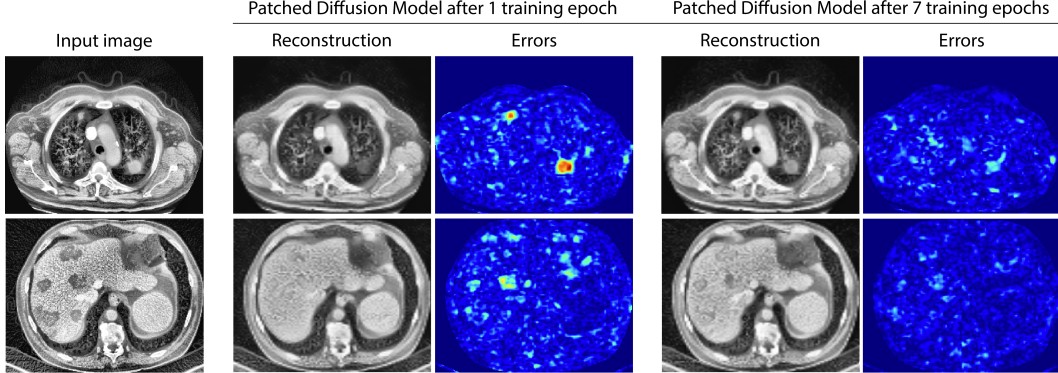

Figure 8: Reconstructions and anomaly maps predicted by Patched Diffusion Model Behrendt et al. (2024) at different epochs. Note that at the beggining of the training (after 1 epoch) it reconstructs healthy regions better than pathologies. However, after 7 epochs, it begins to reconstruct pathologies as well.

## J  FULL-DATA FINE-TUNING

In the fine-tuning experiments (Table 2, Section 4.2), in order to amplify the effect of pretraining we trained all the models on random data subsets consisting of 25 labeled images. Here, in Table 8, we also provide the results of fine-tuning on full training datasets. As anticipated, the benefits of pretraining become substantially smaller. Noticeably, only supervised pretraining, VoCo (Wu et al., 2024), and MAE (Wald et al., 2025) show statistically significant improvements on *some* datasets. In contrast, Screener-based pretraining does not yield gains when abundant labeled data are available.

Table 8: *Full-data* fine-tuning results for Screener and other self-supervised pretrained models. We highlight statistically significant improvements (green) or declines (red) relative to nnUNet trained from random initialization.

| Model | LIDC | MIDRC | KiTS | LiTS |
|---|---|---|---|---|
| nnUNet (random init.) (Isensee et al., 2021) | 0.56 | 0.64 | 0.78 | 0.59 |
| nnUNet (supervised pretrain.) (Huang et al., 2023) | **0.58** ↑ 4% ($p = 0.11$) | 0.65 ↑ 2% ($p = 0.12$) | 0.80 ↑ 2% ($p < 0.01$) | **0.62** ↑ 4% ($p = 0.06$) |
| Grounding DINO (Liu et al., 2023) | 0.43 ↓ 23% ($p < 0.01$) | 0.62 ↓ 3% ($p < 0.01$) | 0.63 ↓ 19% ($p < 0.01$) | 0.49 ↓ 18% ($p < 0.01$) |
| LVM-Med (MH Nguyen et al., 2023) | 0.41 ↓ 25% ($p < 0.01$) | 0.61 ↓ 5% ($p < 0.01$) | 0.62 ↓ 21% ($p < 0.01$) | 0.50 ↓ 15% ($p < 0.01$) |
| Model Genesis (Zhou et al., 2021) | 0.57 ↑ 2% ($p < 0.01$) | 0.62 ↓ 2% ($p < 0.01$) | 0.75 ↓ 4% ($p < 0.01$) | 0.53 ↓ 10% ($p < 0.01$) |
| SwinUNETR (Tang et al., 2022) | 0.52 ↓ 6% ($p < 0.01$) | 0.61 ↓ 4% ($p < 0.01$) | 0.70 ↓ 10% ($p < 0.01$) | 0.50 ↓ 16% ($p < 0.01$) |
| DAE (Valanarasu et al., 2023) | 0.54 ↓ 3% ($p = 0.08$) | 0.63 ↓ 2% ($p < 0.01$) | 0.75 ↓ 4% ($p < 0.01$) | 0.54 ↓ 9% ($p < 0.01$) |
| VoCo (Wu et al., 2024) | 0.57 ↓ 3% ($p = 0.01$) | 0.64 ↑ 0% ($p = 0.93$) | 0.79 ↑ 2% ($p = 0.05$) | 0.60 ↑ 1% ($p = 0.37$) |
| MAE (Wald et al., 2025) | 0.55 ↓ 2% ($p = 0.21$) | **0.66** ↑ 3% ($p < 0.01$) | **0.82** ↑ 5% ($p < 0.01$) | 0.59 ↑ 0% ($p = 0.68$) |
| DenseVICReg | 0.56 ↑ 0% ($p = 0.83$) | 0.63 ↓ 2% ($p < 0.01$) | 0.77 ↓ 1% ($p = 0.06$) | 0.58 ↓ 3% ($p = 0.19$) |
| Screener (ours) | 0.55 ↑ 0% ($p = 0.31$) | 0.65 ↑ 1% ($p = 0.46$) | 0.76 ↓ 2% ($p < 0.01$) | 0.58 ↓ 2% ($p = 0.22$) |

