# OpenReview forum: "Modeling the Density of Pixel-level Self-supervised Embeddings for Unsupervised Pathology Segmentation in Medical CT"
_ICLR.cc/2026/Conference — ICLR 2026 Poster_

### Official Review · Reviewer_otbi · 2025-10-30

**Soundness:** 3
**Presentation:** 4
**Contribution:** 3
**Rating:** 6
**Confidence:** 4

**Summary:**

The authors propose Screener, an unsupervised visual anomaly segmentation (UVAS) method, which leverages dense self-supervised pre-training and masking-invariant dense feature conditioning variables as replacement for positional encodings. The authors train Screener on 30k unlabeled CT-volumes and outperform existing UVAS methods and self-supervised pretraining methods when fine-tuning their method.

**Strengths:**

The method the authors propose, is really nice, as it leverages dense self-supervised pretraining, which is a good choice given their goal of dense downstream applications and their addition of their masking-invariant dense feature conditioning variables as replacement for positional encoding is also well motivated and creative.
Additionally the clarity of the paper is also very high, with the authors explaining the different aspects very well.

**Weaknesses:**

The majority of my criticisms hinge around two key points of the authors paper.

1) The authors claim their method exceeds current self-supervised learning methods. However, the authors don't compare against MAE pre-training, which was shown to be the strongest SSL pre-training for 3D medical image computing in the medical domain in a recent benchmark [1]. In particular the chosen VoCo and SwinUNETR pre-training baselines were shown to be bad for segmentation in general, making this claim not substantiated.

2)  The evaluation protocol of using just 25 training cases is limited. If the authors claim their SSL method to be overall useful for segmentation I would like them to additionally evaluate their pre-training against other pre-training methods on a full-data regime. This is largely, because pre-training methods in general appear to yield performance benefits in small-data regimes, however when more data is available it may not do so anymore. Having this information is crucial for practitioners to know which SSL method to choose given their data availability.


If the authors 1) include MAE pre-training as an additional SSL baseline (when finetuning) and 2) include a finetuning experiment in a  full-data-regime and 3) include a full-length nnU-Net training as reference, I will raise my score further.


[1] Wald, Tassilo, et al. "An OpenMind for 3D medical vision self-supervised learning." Proceedings of the IEEE/CVF International Conference on Computer Vision. 2025.

**Questions:**

Q1: I am not sure if I missed it but how do you get from the anomaly maps to hard predictions as needed for DSC measurement? Is there some thresholding and if so, how is the thresholding done in the unsupervised and supervised setting?

---

> ### Author Response · Authors · 2025-11-21
>
> Dear Reviewer **otbi**, thank you very much for your thoughtful review and for your kind comments on the clarity and creativity of our work. We sincerely appreciate your suggestions regarding MAE and full-data fine-tuning as they are both important and relevant. Below we present the completed MAE experiment and answer your question about the threshold. Full-data fine-tuning and nnUNet experiments will be ready within the next week (they require more GPU-hours).
>
> **Including MAE in Table 2**
>
> Following your recommendation, **we have added MAE in Table 2** as an additional SSL baseline (along with LVM-Med, suggested by Reviewer R1). We used the official pretrained checkpoint from the paper [1] that you have cited (https://huggingface.co/AnonRes/ResEncL-OpenMind-MAE). The updated results are:
>
> | Model                         |      LIDC      |      MIDRC     |      KiTS      |      LiTS      |
> |-------------------------------|:--------------:|:--------------:|:--------------:|:--------------:|
> | nnUNet (random init.)         | 0.21 +- 0.28   | **0.61 +- 0.25**   | 0.41 +- 0.38   | 0.45 +- 0.32   |
> | nnUNet (supervised pretrain.) | **0.29 +- 0.28**   | **0.62 +- 0.22**   | 0.46 +- 0.37   | **0.48 +- 0.30**   |
> | LVM-Med                       | 0.04 +- 0.12   | 0.55 +- 0.24   | 0.16 +- 0.25   | 0.33 +- 0.29   |
> | Model Genesis                 | 0.21 +- 0.29   | 0.59 +- 0.25   | 0.34 +- 0.35   | 0.39 +- 0.31   |
> | SwinUNETR                     | 0.16 +- 0.24   | 0.55 +- 0.23   | 0.19 +- 0.30   | 0.39 +- 0.29   |
> | DAE                           | 0.15 +- 0.22   | 0.58 +- 0.25   | 0.26 +- 0.33   | 0.36 +- 0.29   |
> | VoCo                          | 0.20 +- 0.28   | **0.61 +- 0.23**   | **0.49 +- 0.37**   | **0.49 +- 0.32**   |
> | _MAE_                         | _0.24 +- 0.32_ | **_0.62 +- 0.25_** | _0.35 +- 0.40_ | **_0.49 +- 0.32_** |
> | DenseVICReg                   | 0.22 +- 0.29   | 0.58 +- 0.23   | 0.31 +- 0.33   | 0.44 +- 0.31   |
> | Screener (ours)               | **0.31 +- 0.30**   | **0.62 +- 0.22**   | 0.43 +- 0.35   | **0.48 +- 0.30**   |
>
>
> As can be seen, MAE is slightly superior over Screener on LiTS and MIDRC datasets, while Screener significantly outperforms MAE on LIDC and KiTS datasets. We have updated Table 2 in the manuscript accordingly.
>
> **Including full-data training of pretrained models and nnUNet baseline**
>
> **We will complete and include these experiments within the next week.** Our expectations align with yours: when training on the full public datasets, the gains from pretraining are typically much smaller.
>
> That said, we believe SSL pretraining remains practically valuable in more *incremental* or *active-learning–like* settings, where new data is collected and labeled in small batches of challenging cases. Our 25-case training scenario can be viewed as modeling the *very first iteration* of such a workflow. Understanding how pretraining benefits evolve over subsequent active learning iterations is a very interesting direction, but falls outside the scope of this work.
>
> **Threshold for Dice score calculation**
>
> To obtain binary masks from the anomaly scores, we choose a threshold that maximizes the Dice score on a *small subset of 10 pathological cases* from each test dataset. This threshold is then applied unchanged to the full test set. We have clarified this procedure in the first paragraph of Section 4.1.
>
> Thank you again for your constructive feedback and for giving us the opportunity to expand our experiments. Your suggestions directly strengthen the manuscript, and we greatly appreciate the time you invested in this review.

---

> > ### Comment · Reviewer_otbi · 2025-11-25
> > **Response**
> >
> > Thanks to the authors for including this baseline.
> > I fully believe this inclusion further supports the papers' performance claims and gives it more credibility as a SSL method.
> > In accordance with this, I will raise my rating.

---

> ### Author Response · Authors · 2025-12-01
>
> As promised, we have now completed the full-data fine-tuning experiments:
>
> | Model                         |     LIDC     |     MIDRC    |     KiTS     |     LiTS     |
> |-------------------------------|:------------:|:------------:|:------------:|:------------:|
> | nnUNet (random init.)         | 0.56 +- 0.27 | 0.64 +- 0.24 | 0.78 +- 0.24 | 0.59 +- 0.30 |
> | nnUNet (supervised pretrain.) | **0.58 +- 0.25** | 0.65 +- 0.23 | 0.80 +- 0.22 | **0.62 +- 0.29** |
> | Grounding DINO                | 0.43 +- 0.28 | 0.62 +- 0.24 | 0.63 +- 0.27 | 0.49 +- 0.30 |
> | LVM-Med                       | 0.41 +- 0.28 | 0.61 +- 0.24 | 0.62 +- 0.27 | 0.50 +- 0.29 |
> | Model Genesis                 | 0.57 +- 0.28 | 0.62 +- 0.26 | 0.75 +- 0.27 | 0.53 +- 0.31 |
> | SwinUNETR                     | 0.52 +- 0.27 | 0.61 +- 0.23 | 0.70 +- 0.27 | 0.50 +- 0.30 |
> | DAE                           | 0.54 +- 0.28 | 0.63 +- 0.24 | 0.75 +- 0.25 | 0.54 +- 0.29 |
> | VoCo                          | 0.57 +- 0.28 | 0.64 +- 0.24 | 0.79 +- 0.23 | 0.60 +- 0.29 |
> | MAE                           | 0.55 +- 0.26 | **0.66 +- 0.24** | **0.82 +- 0.21** | 0.59 +- 0.31 |
> | DenseVICReg                   | 0.56 +- 0.28 | 0.63 +- 0.23 | 0.77 +- 0.25 | 0.58 +- 0.31 |
> | Screener (ours)               | 0.56 +- 0.26 | 0.65 +- 0.23 | 0.76 +- 0.24 | 0.58 +- 0.29 |
>
> As anticipated, the benefits of all the pretrained models are substantially smaller in the full-data regime compared to the low-data results in Table 2 (can be found in our previous response and in the paper). In particular, Screener-based pretraining does not yield any gains in full-data regime, which is consistent with the broader observation that many SSL methods provide their strongest advantages when training data are scarce.
>
> We have added these results and discussion to Section 4.2 and Appendix J.

---

### Official Review · Reviewer_NAFn · 2025-10-31

**Soundness:** 3
**Presentation:** 3
**Contribution:** 3
**Rating:** 6
**Confidence:** 4

**Summary:**

This paper proposes Screener, a self-supervised anomaly segmentation framework for 3D CT. The method combines dense self-supervised descriptor learning, masking-invariant conditioning embeddings, and density modeling to detect abnormal patterns. The model trains on over 30,000 unlabeled CT volumes. Screener is tested on four public CT benchmarks under both unsupervised and fine-tuning settings.

**Strengths:**

The problem is clearly motivated, as a supervised method cannot capture all distributions of pathologies.

The integration of dense SSL for descriptors and SSL-learned conditioning embeddings looks novel.

The proposed model is tested on four public CT benchmarks under both unsupervised and fine-tuning settings. Strong and consistent performance is observed.

**Weaknesses:**

While the method is well-executed, the contribution may appear incremental as the dense SSL for voxel embeddings and conditioning is a straightforward extension of VICReg and conditioning replacing sin-cos encodings is conceptually natural. The paper would benefit from clearer articulation of why condition embeddings are fundamentally more informative than APE or sin-cos encodings beyond empirical gains.

The proposed method looks closely related to an existing method [1]. It would be beneficial if the authors acknowledge and contrast with the prior work.

The method produces per-voxel anomaly scores. A threshold is necessary to perform the final segmentation. It is unclear how this threshold is determined.

References:

[1] Seince, Maxime, Loı̈c Le Folgoc, Luiz Facury De Souza, and Elsa Angelini. "Dense Self-Supervised Learning for Medical Image Segmentation." In Medical Imaging with Deep Learning, pp. 1371-1386. PMLR, 2024.

**Questions:**

1. Are there reasons why condition embeddings are fundamentally more informative than APE or sin-cos encodings?

2. How is the threshold determined to obtain the segmentation?

---

> ### Author Response · Authors · 2025-11-21
>
> Dear Reviewer **NAFn**, thank you for taking time to review our submission and for recognizing the novelty of our method. We greatly value your feedback and have done our best to address your concerns below.
>
> **Contribution, novelty and comparison with prior works**
>
> As stated in Section 3, novelty of our work lies not in dense SSL *per se*, but in its **integration into the density-based UVAS framework**, enabling voxel-level SSL embeddings to be modeled probabilistically for *unsupervised anomaly segmentation*. Our key contribution (stated in Section 1) is demonstrating that **conditional density modeling of voxel-level SSL embeddings yields state-of-the-art unsupervised anomaly segmentation in CT**, outperforming existing UVAS methods by a significant margin. This distinguishes our work from VICRegL and from [1], which use dense SSL representations *only as features for supervised segmentation*, without any density modeling or anomaly detection component.
>
> Importantly, we also show that **fine-tuning a UNet distilled from our anomaly segmentation model outperforms fine-tuning of dense SSL features alone** (Screener vs. DenseVICReg in Table 2), demonstrating that our anomaly-driven pretraining is complementary to conventional dense SSL.
>
> **Why are our condition embeddings more informative than APE and sin-cos embeddings?**
>
> Our voxel-wise condition embeddings are **learnable, image-driven 32-dimensional features** trained to maximize voxel discriminability while being invariant to random image masking. This design allows the model to encode nuanced contextual cues (organ identity, tissue type, patient-level geometry) while suppressing pathological information.
>
> In contrast, sin-cos positional embeddings *are not learnable and do not encode visual content*. Voxels at the same spatial coordinates in different patients have identical embeddings, even if they correspond to entirely different organs or structures.
>
> APE embeddings explicitly encode anatomical relative distances, which clusters anatomically distant organs well. However, *adjacent but distinct organs* (e.g., the hepatic dome vs. lower right lung) *remain poorly separable because their APE vectors are inherently close by design*. Thus, APE provides coarse anatomical grouping but insufficient fine-grained contextual separation.
>
> To empirically demonstrate that our condition embeddings encode more useful contextual information, **we provide an additional probing experiment**. We train the same lightweight segmentation head (conv1×1×1 → BN → ReLU → conv1×1×1 → BN → ReLU → conv1×1×1) on top of frozen condition embeddings, APE, and sin-cos embeddings, and evaluate performance on *liver* segmentation (LiTS):
>
> | Model                        | LiTS (liver) |
> |------------------------------|:------------:|
> | Condition model probe        | **0.82 +- 0.14** |
> | APE probe                    | 0.62 +- 0.13 |
> | Sin-cos pos. embed. probe    | 0.36 +- 0.14 |
>
> Condition embeddings separate liver vs. non-liver voxels substantially better than both APE and sin-cos encodings. We have updated the manuscript by including this probing experiment in Appendix D.
>
> **Threshold for segmentation and Dice score calculation**
>
> To obtain binary masks from the anomaly scores, we choose a threshold that maximizes the Dice score on a *small subset of 10 pathological cases* from each test dataset. This threshold is then applied unchanged to the full test set. We have clarified this procedure in the first paragraph of Section 4.1.
>
> Thank you again for your constructive feedback and for helping us improve the clarity and completeness of the manuscript.

---

### Official Review · Reviewer_Waas · 2025-10-31

**Soundness:** 3
**Presentation:** 3
**Contribution:** 3
**Rating:** 6
**Confidence:** 4

**Summary:**

The paper presents Screener, a fully self-supervised framework for unsupervised pathology segmentation in 3D CT. It reframes pathology detection as unsupervised visual anomaly segmentation (UVAS), under the assumption that pathological regions are statistically rare. The method enhances density-based UVAS by (i) learning dense voxel-level self-supervised descriptors tailored to CT data, and (ii) introducing masking-invariant contextual embeddings as conditioning variables within a conditional density model. The models are trained on 30k unlabeled CT volumes. Screener achieves state-of-the-art results on four public CT datasets (LIDC, MIDRC, KiTS, LiTS). When distilled into a single UNet and fine-tuned with limited labeled data, it matches or surpasses medical SSL baselines.

**Strengths:**

Reviewer acknowledges the following contributions:

 **Addresses an important medical challenge**: The paper tackles the critical and realistic problem of detecting all pathological findings in 3D CT without requiring pixel-level annotation, which is a limitation that hinders clinical deployment of current supervised methods.

 **Interesting and well-motivated technical idea**:

- The proposed use of self-supervised learning for both the descriptor and conditioning models is conceptually appealing. It enables the system to model normal anatomy and contextual expectations directly from large unlabeled data, thereby detecting deviations that correspond to abnormalities.

- Furthermore, the integration of dense SSL with masking-invariant conditioning is simple but effective. It allows a simple Gaussian density model to perform on par with more complex normalizing flows, indicating that meaningful contextual embeddings can simplify anomaly modeling.

**Broad evaluation**: Experiments across four diverse datasets demonstrate that Screener generalizes well across organs and pathologies. The unsupervised anomaly segmentation results are significant, with clear ablation studies supporting each component.

 **Clear presentation**: The paper is clearly written, logically organized, and easy to follow, making complex ideas accessible to both machine learning and medical imaging audiences.

**Weaknesses:**

I found the following weaknesses in the current manuscript:

- **Core conceptual clarity**: The motivation for why optimizing the **conditional density model** $q_{\theta}(y|c)$. helps detect abnormal regions, is not fully articulated. Since this is a central concept, further clarification and discussion would strengthen the theoretical grounding. For instance, authors need to provide a visualization of normal pixels near the abnormal regions and show how the heatmap behaves to provide insights.

- **Experimental organization**: The unsupervised experiments could be better structured to highlight the benefits of dense SSL. For instance, baselines should be grouped into (i) image-level SSL models (e.g., autoencoder) as well as adding SOTA models like LVM-Med [1] (developed for ResNet-50 and suitable with U-Net) and (ii) dense SSL models (the current author compared), to clearly demonstrate that dense SSL drives the improvements.

- **Supervised performance gap** – While competitive, the fine-tuned Screener in Table 2 lags behind certain SOTA supervised or SSL-pretrained models. The authors could explore initializing the distilled UNet with a pretrained medical backbone (e.g., RadImageNet or LVM-Med to enhance feature representations.

- **Presentation detail** – Some result table formats (e.g., Tables 1–3) could be reformatted and polished for a more professional appearance and readability.

[1] Lvm-med: Learning large-scale self-supervised vision models for medical imaging via second-order graph matching, NeurIPS 2023.

[2] RadImageNet: an open radiologic deep learning research dataset for effective transfer learning." Radiology: Artificial Intelligence

**Questions:**

- In the self-supervised learning settings, besides the medical models, can authors examine the performance of the generalized model, for e.g., the Grounding Dino technique, which can retrieve objects given prompt input and apply some threshold techniques to filter noise [3]? Incorporating such experiments is interesting and highlights the benefit or potential weakness of the current strategy.

-  It would be great to see the advanced performance by applying the U-Net with pre-trained medical models during the distillation process, aiming to bridge the gap between current performance and other SOTA.


[3] Grounding Dino 1.5: Advance the" edge" of open-set object detection.

---

> ### Author Response · Authors · 2025-11-21
>
> Dear Reviewer **Waas**, thank you for your thoughtful review and constructive feedback. We appreciate your recognition of our contributions and address your comments in detail below.
>
> **Core conceptual clarity: how and why condition model helps?**
>
> Our intuition – introduced in Section 3, and empirically validated in Table 3 – is that an effective conditioning variables should (i) exclude information about presence / absence of pathologies, while (ii) *capturing as much contextual information as possible* (e.g., anatomical region, tissue type, patient-level characteristics etc.). This combination simplifies conditional density modeling and sharpens anomaly detection.
>
> To make this intuition explicit, **we conducted an additional probing experiment** comparing the amount of pathology-related information retained by the descriptor model versus the condition model. We trained a lightweight segmentation head (conv1×1×1 → BN → ReLU → conv1×1×1 → BN → ReLU → conv1×1×1) on top of frozen descriptor vs. condition embeddings and evaluated Dice scores on four pathology segmentation tasks:
>
> | Model                         |     LIDC     |     MIDRC    |      KiTS     |     LiTS     |
> |-------------------------------|:------------:|:------------:|:-------------:|:------------:|
> | Descriptor model probe        | **0.19 +- 0.23** | **0.40 +- 0.19** | **0.17 +- 0.18**  | **0.25 +- 0.23** |
> | Condition model probe         | 0.00 +- 0.00 | 0.27 +- 0.16 | 0.06 +- 0.10  | 0.01 +- 0.03 |
>
> Condition model probing yields substantially lower Dice scores (p-value < 1e−10 for all datasets). For lung cancer and liver tumors, scores are essentially zero, confirming that pathology-specific information is effectively removed. For pneumonia and kidney tumors, non-zero Dice arises largely because these pathologies occupy substantial portions of the corresponding organs; the segmentation head tends to predict the whole organ, producing non-zero Dice scores.
>
> To demonstrate that our condition model indeed captures *anatomical* information better than baselines, we also probe it on *liver* segmentation task on LiTS:
>
> | Model                        | LiTS (liver) |
> |------------------------------|:------------:|
> | Condition model probe        | **0.82 +- 0.14** |
> | APE probe                    | 0.62 +- 0.13 |
> | Sin-cos pos. embed. probe    | 0.36 +- 0.14 |
>
> As seen, condition model embeddings separate liver vs. non-liver voxels substantially better than APE and sinusoidal positional encodings.
>
> We have updated the manuscript by including these probing experiments in Appendix D.
>
> **Experimental organization: adding and grouping baselines in Table 2**
>
> We have updated the manuscript by **adding LVM-Med (following your suggestion) and MAE (following Reviewer **otbi**’s suggestion) as two more baselines in Table 2**. We will also include Grounding-DINO baseline within the next week. We organize baselines in Table 2 into four groups: “from scratch”, “supervised pretraining”, “self-supervised pretraining” and our “anomaly-driven pretraining”. Grounding-DINO will be included in a separate “non-medical pretraining” group.
>
> **Supervised performance gap and combining Screener distillation with supervised pretraining**
>
> We clarify that Screener underperforms supervised pretraining *only on KiTS21*, while even *slightly outperforming* it on LIDC. We agree that a two-stage strategy (supervised pretraining + Screener distillation) may offer further improvements. However, distillation requires ~5 days of large-scale training, and due to GPU constraints we are unable to run this experiment during the rebuttal period and would like to leave it for future work.
>
> **Presentation detail: formatting Tables 1-3**
>
> We have corrected citation formatting in Tables 1-3 and align columns in Table 2. We would be happy to incorporate any additional formatting suggestions.
>
> Thank you again for your time, effort, and constructive insights. We believe the revisions and additional experiments substantially strengthen the manuscript and clarify the core ideas.

---

> > ### Comment · Reviewer_Waas · 2025-11-25
> > **Response to authors**
> >
> > Thanks to the authors for responding to my concern. Regarding the newly added baselines, it's particularly noteworthy to me that LVM-Med's performance is 0.04, which is an outlier compared to the others. Can authors please verify the hyperparameters and other configurations, such as norm gradients, for this dataset and others? In general, what I mean is that an extremely low value of the LVM-Med model can raise concern about whether the authors conducted the right training setting.
> >
> > Also, please highlight your changed text in color for easier follow.

---

> ### Author Response · Authors · 2025-11-26
>
> Dear Reviewer **Waas**, thank you for your attentive follow-up and for raising this important point regarding the LVM-Med baseline.
>
> We would like to clarify that **LVM-Med is built on a 2D ResNet-50 encoder**, and therefore operates strictly with **2D convolutions**. To apply it to 3D CT volumes, we must process the scans *slice-wise*, without any native 3D spatial perception. In contrast, **all other baselines, along with Screener, use 3D convolutional or 3D attention–based architectures**, which are well-known to offer substantial advantages on volumetric medical segmentation tasks.
>
> This architectural mismatch is particularly impactful on LIDC, where the task is pulmonary nodule segmentation. **Many nodules are impossible to distinguish from lung vessels when viewed on individual 2D slices**, as their appearance can be nearly identical without 3D context. As a result, LVM-Med performs particularly poorly on this dataset, yielding the observed Dice score of 0.04. On the other three datasets, where purely 2D context is less misleading, LVM-Med achieves more reasonable scores (0.55, 0.16, 0.33), though it remains consistently inferior to 3D architectures.
>
> Regarding the correctness of our training procedure, **all the baselines are run using a unified, shared training pipeline**, the overall pattern of results across models is consistent, and no anomalies were observed during training. To enable transparent verification, we provide our **anonymous code repository: https://anonymous.4open.science/r/screener-35EE/** (supervised fine-tuning pipeline can be found in `src/screener/models/supervised.py` and `src/screener/datamodules/supervised.py` modules). We also provide a **tensorboard screenshot** with LVM-Med, distilled Screener and nnUNet training and validation logs: **https://pdfhost.io/v/TYvaZ8D8jq_tb_screenshot**. Note that LVM-Med actually overfits to the training set, but fails to generalize to the val set (most likely for the reasons discussed in the previous paragraph).
>
> Finally, following your recommendation, all changes in the manuscript are now highlighted in blue for easier review.
>
> Thank you again for your careful consideration and constructive feedback.

---

> > ### Comment · Reviewer_Waas · 2025-11-27
> >
> > Dear Authors
> >
> > > This architectural mismatch is particularly impactful on LIDC, where the task is pulmonary nodule segmentation. Many nodules are impossible to distinguish from lung vessels when viewed on individual 2D slices, as their appearance can be nearly identical without 3D context. As a result, LVM-Med performs particularly poorly on this dataset, yielding the observed Dice score of 0.04. On the other three datasets, where purely 2D context is less misleading, LVM-Med achieves more reasonable scores (0.55, 0.16, 0.33), though it remains consistently inferior to 3D architectures.
> >
> > Yes, this explanation makes sense to me. Probally, you should a small note about the mismatch architecture in the caption table to avoid potential reader's concerns. Thank you and good luck!

---

> ### Author Response · Authors · 2025-11-30
>
> As promised, we have now included Grounding DINO as non-medical self-supervised baseline in Table 2:
>
> | Model                         |      LIDC      |      MIDRC     |      KiTS      |      LiTS      |
> |-------------------------------|:--------------:|:--------------:|:--------------:|:--------------:|
> | nnUNet (random init.)         | 0.21 +- 0.28   | **0.61 +- 0.25**   | 0.41 +- 0.38   | 0.45 +- 0.32   |
> | nnUNet (supervised pretrain.) | **0.29 +- 0.28**   | **0.62 +- 0.22**   | 0.46 +- 0.37   | **0.48 +- 0.30**   |
> | _Grounding DINO_              | _0.09 +- 0.19_ | _0.58 +- 0.23_ | _0.19 +- 0.26_ | _0.33 +- 0.29_ |
> | LVM-Med                       | 0.04 +- 0.12   | 0.55 +- 0.24   | 0.16 +- 0.25   | 0.33 +- 0.29   |
> | Model Genesis                 | 0.21 +- 0.29   | 0.59 +- 0.25   | 0.34 +- 0.35   | 0.39 +- 0.31   |
> | SwinUNETR                     | 0.16 +- 0.24   | 0.55 +- 0.23   | 0.19 +- 0.30   | 0.39 +- 0.29   |
> | DAE                           | 0.15 +- 0.22   | 0.58 +- 0.25   | 0.26 +- 0.33   | 0.36 +- 0.29   |
> | VoCo                          | 0.20 +- 0.28   | **0.61 +- 0.23**   | **0.49 +- 0.37**   | **0.49 +- 0.32**   |
> | MAE                           | 0.24 +- 0.32   | **0.62 +- 0.25**   | 0.35 +- 0.40   | **0.49 +- 0.32**   |
> | DenseVICReg                   | 0.22 +- 0.29   | 0.58 +- 0.23   | 0.31 +- 0.33   | 0.44 +- 0.31   |
> | Screener (ours)               | **0.31 +- 0.30**   | **0.62 +- 0.22**   | 0.43 +- 0.35   | **0.48 +- 0.30**   |
>
> Its relatively low performance is expected: like LVM-Med, Grounding DINO is 2D model, operating on individual slices without access to the full 3D image context that is crucial for volumetric CT segmentation. We have added a note about the architectural mismatch between 2D vs. 3D models in Table 2 caption.

---

### Author Response · Authors · 2025-12-03
**Author Summary for Area Chair**

Dear Area Chair and Reviewers,

Thank you for reviewing our submission and for the constructive feedback. Below is a brief summary of the reviews and the ensuing discussion.

**Recognized strengths**

R1, R2 emphasized that framing pathology segmentation as an unsupervised anomaly segmentation problem **is well-motivated and important**.

All reviewers recognized the **method’s novelty**:
- R1: “Interesting and well-motivated technical idea… the use of self-supervised learning for both the descriptor and conditioning models is conceptually appealing.”
- R2: “The integration of dense SSL for descriptors and SSL-learned conditioning embeddings looks novel.”
- R3: “The method the authors propose, is really nice...” and the masking-invariant conditioning embeddings are “well motivated and creative.”

R1, R2 highlighted **broad evaluation and strong empirical results**.

Last but not least, R1 and R3 praised the **clear presentation**, noting that the paper “makes complex ideas accessible”.

**Reviewers’ concerns & how we addressed them**

R1 and R2 asked for deeper intuition regarding why our learned condition embeddings are beneficial for anomaly detection. In response, we **added probing experiments (Appendix D)** explicitly showing that our condition embeddings do not preserve pathology-related signals, but, at the same time, capture richer contextual features than APE or sin–cos embeddings. This combination simplifies conditional density modeling and sharpens anomaly detection.

R2 requested comparison to prior work on dense SSL for *supervised* medical image segmentation. We clarified that our contribution lies not in dense SSL *per se*, but in *modeling the probability density in the space of dense SSL embeddings for unsupervised anomaly segmentation*.

Following suggestions from R1 and R3, we **added LVM-Med, MAE, and Grounding-DINO** to the low-shot fine-tuning experiments **in Table 2**. After we added the MAE baseline, **R3 raised their score to 8 (as explicitly stated in one of their comments)**, but the score was subsequently reverted following the ICLR incident.

Following R3’s request, we also **added full-data fine-tuning results in Table 8, in Appendix J**.

**Final notes**

Across the discussion, all reviewers acknowledged the importance of the problem, method's novelty, extensive experiments, strong results and paper's clarity. All major concerns have now been addressed and reflected in the revised manuscript.

Thank you again for your time and for helping us strengthen the paper.

---

### Meta-Review · Area_Chair_MpMq · 2025-12-23

**Summary:**

This paper introduces a well‑motivated framing of pathology segmentation as an unsupervised anomaly segmentation problem, which both R1 and R2 emphasized as important and impactful. All reviewers acknowledged the novelty of the method: R1 praised the “interesting and well‑motivated technical idea” and highlighted the conceptual appeal of leveraging self‑supervised learning for both the descriptor and conditioning models; R2 noted the novelty in integrating dense SSL for descriptors with SSL‑learned conditioning embeddings; R3 described the approach as “really nice” and commended the masking‑invariant conditioning embeddings as “well‑motivated and creative.” Given the combination of a novel and well‑justified technical approach, broad and strengthened empirical evaluation, and clear, accessible presentation, alongside the fact that reviewers’ key concerns have been substantively addressed in the rebuttal, the AC recommends acceptance of this paper.

**Reviewer Concerns:**

Reviewer concerns were addressed during the rebuttal process.

**Reviewer Scores:**

Based on the reviewers’ comments and the authors’ responses, the AC believes that all four reviewers are likely to maintain a positive rating.

---

### Decision · Program_Chairs · 2026-01-26

Accept (Poster)